# Ecological Groups of Coleoptera (Insecta) as Indicators of Habitat Transformation on Drained and Rewetted Peatlands: A Baseline Study from a Carbon Supersite, Kaliningrad, Russia

**DOI:** 10.3390/insects15050356

**Published:** 2024-05-15

**Authors:** Vitalii Alekseev, Maxim Napreenko, Tatiana Napreenko-Dorokhova

**Affiliations:** Scientific and Educational Centre for Environmental Geology and Maritime Management, Immanuel Kant Baltic Federal University, Kaliningrad 236016, Russia; viialekseev@kantiana.ru (V.A.); tnapdor@atlantic.ocean.ru (T.N.-D.)

**Keywords:** coleoptera, indicator species assemblage, disturbed peatlands, rewetting, Baltic Sea region, checklist

## Abstract

**Simple Summary:**

On the modern climate agenda, drained peatlands are considered ‘hotspots’ that are responsible for a significant share of anthropogenic greenhouse gas emissions. The solution to this problem is usually associated with peatland rewetting, which aims to restore natural mire communities and their peat accumulation functions. The process of habitat recovery on both drained and rewetted peatlands requires the monitoring of biodiversity and ecosystem structures, with particular attention being paid to specific groups of organisms that may have an indicator value. Among these, some groups of beetles (Coleoptera)—as this order is the most species-rich on Earth—can be relatively indicative of specific environmental conditions, including different peatland habitats. Our studies provide baseline data on the beetle fauna of a drained peatland in the Kaliningrad region of Russia, designated for rewetting as part of the Carbon Measurement Supersite Programme. The recorded assemblage of 281 beetle species from 41 families differs significantly from the natural raised bog fauna. An analysis of the species composition enabled us to classify the beetle assemblage into specific groups according to their ecological relationships and peat bog biotope fidelity. Based on these results, we have identified the beetle indicator species for further monitoring the environmental changes during peatland rewetting.

**Abstract:**

A total of 281 coleopteran species from 41 families were recorded from different sites of an abandoned cut-over peatland designated as the Carbon Measurement Supersite in Kaliningrad Oblast. This beetle assemblage is considered a baseline (pre-impact) faunal assemblage for further investigations during the ‘before–after’ (BA) or ‘before–after control-impact’ (BACI) study on a peatland that is planned to be rewetted. The spontaneously revegetated peatland has a less specialised beetle assemblage than at an intact raised bog. Tyrphobiontic species are completely absent from the peatland, while some tyrphophiles (5.3% of the total beetle fauna) are still found as remnants of the former raised bog communities. The predominant coenotic coleopteran group is tyrphoneutral generalists from various non-bog habitats (72.9%). The species composition is associated with the vegetation structure of the disturbed peatland (fragmentary *Sphagnum* cover, lack of open habitats, and widespread birch coppice or tree stand), which does not correspond to that of a typical European raised bog. The sampled coleopteran assemblage is divided into several relative ecological groups, whose composition and peculiarities are discussed separately. Possible responses to the rewetting measurements in different coleopteran groups are predicted and briefly discussed. A complex assemblage of stenotopic peatland-specialised tyrphophiles (15 spp.) and the most abundant tyrphoneutral generalists (31 spp.) were assigned as indicators for the environmental monitoring of peatland development.

## 1. Introduction

Peatlands are a unique type of ecosystem, occupying a relatively small fraction of the Earth’s land area, but playing an important role in the global carbon cycle as huge carbon sinks [1]. On the other hand, these communities provide a wide range of structural, hydrological, and vegetational diversity, especially if we consider anthropogenically modified peatlands. Although intact and disturbed peatlands often appear to be similar in terms of their physiognomy and ecology, they can have contrasting environmental relevance, playing a positive role in carbon accumulation and specific biodiversity conservation (natural peatlands) or being hotspots for greenhouse gas emissions and habitat loss (disturbed peatlands).

The issue of habitat transformation is often raised in the context of rewetting projects on disturbed peatlands, which aim to restore natural mire communities and their peat accumulation functions for GHG sequestration. However, peatland restoration usually does not return ecosystems to their original state, but often leads to the formation of completely new communities with unclear consequences at the local and global scales [2] or to a loss of heterogeneity and consequently to a loss of species richness [3,4]. In this case, the recovery process requires the monitoring of the widest possible range of ecosystem components at all stages of ecosystem development. The characteristic species or species groups of mosses and vascular plants are traditionally of great importance in these investigations when applied to the disturbed peatlands. However, there are other specific groups of organisms that can have an important indicator value during monitoring surveys.

Beetles (Coleoptera) are the most species-rich insect order on Earth; they have diverse and important ecosystem functions and are well represented in different freshwater and terrestrial ecosystems, as well as in different peatland habitats. Some groups of these insects can be relatively indicative of highly specific environmental conditions [5] and their changes over the course of succession within an ecosystem. The historical development and nature of individual bogs are reflected in their insect assemblages [6], and monitoring Coleoptera could be interesting in the context of environmental change. Studies of coleopteran fauna and ecology in various bog habitats and successional stages of different peatlands are not rare per se, but the knowledge of their biota is insufficient and very uneven [7]. Much of the coleopteran research has been carried out on the biodiversity of the most natural, relic, and pristine bogs with a focus on epigeic or water beetles (e.g., [8,9,10,11,12,13,14,15,16,17,18,19]). Data on other groups of peatland beetles (e.g., [20,21,22]) or comprehensive studies of the entire fauna (e.g., [23]) are generally scarce. Faunal surveys of severely disturbed and drained peatlands with degraded bog fauna [24,25] or the fauna of rewetted peatlands after their artificial restoration [26] are under-represented. With respect to Coleoptera, only one untouched peat bog within the present territory of Kaliningrad Oblast is relatively well studied [9,11,23,27].

In this context, obtaining data on different groups of Coleoptera in a drained peatland seems to be an interesting and relevant part of comprehensive investigations at the Rossyanka Carbon Supersite, as they could be involved in the monitoring programme on sites designated for rewetting in correlation with the other indicator groups. The results of our peatland study are promising and interesting due to the fragmentary knowledge of the coleopteran fauna of peatlands in Kaliningrad Oblast and the paucity of specialised literature.

The objectives of our study were (1) to report pre-impact data on the coleopteran taxonomic diversity in an abandoned cut-over peatland in Kaliningrad Oblast, Russia; (2) to determine the structure of the beetle assemblage in terms of species arrangement within ecological groups and peat bog affinity categories; (3) to identify indicator species and species assemblages relevant for monitoring environmental changes during further peatland rewetting.

## 2. Materials and Methods

The investigated territory (Figure 1 and Figure 2), namely the Vittgirrensky peatland (54.799° N, 21.658° E), with a total area of 122 ha, is located in the Slavsk administrative district of Kaliningrad Oblast southwards from the extinct village of Wittgirren (=Weißenbruch), 18 km SE of Bol’shakovo (=Groß Skaisgirren/Kreuzingen), and 6 km NE of the Vysokoe (=Popelken/Markthausen). The Vittgirrensky peatland is a ditched, drained, and heavily human-transformed raised bog. The edge zone of this bog was drained at the end of the 19th century and peat was mined by hand cutting [28]. In the late 1970s and 1980s, the peatland was completely drained by a combined (open and closed) drainage system. The area was intensively used for commercial peat extraction by the milling method during the period 1980–1996. Later, the Vittgirrensky peat cutting area was abandoned after the peat extraction ceased. Judging by burnt tree trunks and charcoal residues in the uppermost peat layers, the area was the subject of local surface fires. In 2021, the area was declared as a carbon supersite (‘Rossyanka’) to estimate and monitor greenhouse gas fluxes as well as study biodiversity and test rewetting technologies for fire prevention, climate change mitigation, and habitat restoration.

The study area is located in a region with a temperate climate with an average annual precipitation of 750 mm and an average annual temperature of +7.8 °C. Our own measurements showed the following average monthly temperatures and average monthly liquid precipitation on the peatland area during the year 2023: +12.1 °C and 7.44 in May, +16.7 °C and 86.4 mm in June, +17.3 °C and 81.84 mm in July, +19.2 °C and 119.04 mm in August, +16.4 °C and 36.0 mm in September, +8.6 °C and 89.28 mm in October, and +2.2 °C and 36 mm in November. The frost-free period was 180 days.

At present, the peatland is covered by regrowth communities that have developed after the destruction of primary bog vegetation [28]. Vegetation cover is more diverse in areas with stable soil moisture (remnants of bog and wetland communities) and along the central road (ruderal plant community). The current dominant arboreal species on the Vittgirrensky peatland is birch (*Betula pendula* and *B. pubescens*), but a small admixture of *Populus tremula*, *Salix* spp. (at least 8 species), *Sorbus aucuparia*, and *Pinus sylvestris* is also present. The birch coppice and regrowth form different mosaic communities (with *Calluna*, *Ledum*, *Phragmites*) of mainly semi-open character with the oldest trees being about 30 years old (Figure 2). Bare-peat sites, fen-like communities (*Juncus*, *Eriophorum*, *Carex*), and birch thickets can be also found in the peatland. Detailed maps and comprehensive data on vegetation and soils were given in recent publications [28,29,30]. The Vittgirrensky peatland is situated within an intensively used open agricultural landscape in relative isolation from faunal interchanges with the pristine or semi-natural bog habitats for a long time. The nearest intact raised bog Bol’shoye (former Mupiau/Palmbruch, with a total area of about 600 ha), is located 7 km southwest (54,735° N, 21,615° E).

The material presented in this paper was collected by the first author (V.A.) between April and November 2023. Adult Coleoptera were collected using the following methods and equipment:(1)Sweeping with entomological sweep nets during 7 days of field surveys (07.04.2023; 13.05.2023; 13.06.2023; 13.07.2023; 13.08.2023; 17.09.2023; 13.10.2023) in different terrestrial and aquatic biotopes of the study area;(2)Visual search of vegetation, soil, fungi, as well as under pieces of wood and bark with hand collection during 8 field survey days (07.04.2023; 13.05.2023; 13.06.2023; 13.07.2023; 13.08.2023; 17.09.2023; 13.10.2023; 17.11.2023) in different terrestrial biotopes of the study area;(3)Pitfall traps without bait (the 250 cm^3^ plastic containers with 9% acetic acid as a preservative liquid, placed in lines of 10 traps with a distance between traps of about 1 m) in seven-month periods between field surveys (07.04–13.05.2023; 13.05–13.06.2023; 13.06–13.07.2023; 13.07–13.08.2023; 13.08–17.09.2023; 17.09–13.10.2023; 13.10–17.11.2023) in 3 terrestrial biotopes of the studied area (Figure 2 and Figure 3). The trap lines were placed in: (1) the N edge of the peatland (sun-exposed site with *Calluna*, *Eriophorum*, and *Betula* coppice), (2) the SW edge of the peatland (dense birch stand); and (3) the central part of the peatland (sun-exposed site with partially bare peat, *Campylopus introflexus,* and *Betula* regrowth). Several containers were effective in attracting carrion beetles (Silphidae, Cholevinae) after the accidental capture of shrews, voles, and lizards in the originally unbaited trap.

The larval stages of the beetles were ignored, except for three species that are easily distinguishable in the wild (*Endomychus coccineus*, *Harmonia axyridis*, and *Schizotus pectinicornis*). Voucher specimens for all the species that are difficult to identify were deposited in the private collection of the first author (Kaliningrad, Russia; abbreviation: CVIA). Taxa common in Kaliningrad Oblast and identifiable in the field were recorded without collecting specimens. The specimens were identified by the first author using the identification keys [31,32,33,34]. The order used in the checklist is systematic for families and is given according to the recent classification [35]. Genera and species collected or observed in the field are arranged alphabetically (by scientific name).

A rough estimation and A.F.O.R. scale was used to measure local abundance with the following values (see also Appendix A):(1)Abundant (A)—species observed or sampled by ≥10 specimens/day or species pitfall-trapped by ≥10 specimens/month;(2)Frequent (F)—species observed or sampled by 5–9 specimens/day or species pitfall-trapped by ≥5–9 specimens/month;(3)Occasional (O)—species observed or sampled by 2–4 specimens/day or species pitfall-trapped by 2–4 specimens/month;(4)Rare (R)—species observed or sampled by single specimen/day or species pitfall-trapped by single specimen/month.

The structure of the species profile in the checklist includes the following: full species name (maximum local abundance estimated using the A.F.O.R. scale); the sampling method; the date or period of capture; location; ecological notes (if any); and ecological category of the species. Three main sampling methods are indicated as ‘net’ (for sweeping with entomological hand net in water or through vegetation), ‘visual’ (for hand collection) or ‘pitfall’ (for pitfall traps). Approximate sampling locations are abbreviated as follows: C—central part (core peatland area, approximately restricted by corresponding quarters), E—eastern quarter of total area, N—northern, S—southern, SE—southeastern, SW—southwestern, W—western. The basic ecological category for each species was chosen from four options according to the classification of peat bog entomofauna by Spitzer and Danks [6]. The possible groups are *tyrphobiontic* species (found only in bogs, not in the study area), *tyrphophilous* species (characteristic of bogs but not restricted to them), *tyrphoneutral* species (resident in bogs and peatlands, but also in other habitats), and *tyrphoxenous* species (non-resident vagrants or erratics, unable to form stable perennial populations in bogs and peatlands). The inhabitants of roadside ruderal plant communities were considered tyrphoxenous.

## 3. Results and Discussion

The examination and identification of the material from the study area (samples from 2023) resulted in a list of 281 species (41 families) was compiled. Among the Coleoptera collected, 231 species (82%) are terrestrial in the adult stage and 50 species (18%) are aquatic. A total of 166 coleopteran species were collected by sweep netting in water or on vegetation (144 species exclusively by this method); 104 species were sampled using terrestrial pitfall traps (83 species exclusively by this method); and 41 species were recorded by other methods (21 species registered only in this way). In total, 30 species were recorded using two or more methods.

A complete preliminary list of the beetle species found in the study area during the 2023 sampling season is given in Appendix A.

### 3.1. Taxonomic Structure and Abundance

In order to assess the taxonomic structure of the beetle fauna in the study area, we present our results in comparison with data from a nearly intact large (ca. 2500 ha) raised bog—Zehlau—located 50 km southwards (Figure 1), which is considered a reference site due to the fact that several entomological inventories were carried out there during the last century. Skwarra [23] collected 332 coleopteran species in the Zehlau raised bog between 1920 and 1926. Främbs et al. [11] sampled 36 species of the family Carabidae in 1994 in the Zehlau bog and its nearest vicinity. Biesiadka and Moroz [9] reported 43 species of the family Dytiscidae. The brief comparison of our results with the peatland fauna recorded by Skwarra [23], Främbs et al. [11], and Biesiadka and Moroz [9] is possible, although it can only be performed with reservations (different structural complexity and set of habitats, different studied area, different sampling methods, etc.).

The faunistic core in both peatlands consists of the same most species-rich families (≥10 species): Cantharidae, Carabidae, Chrysomelidae, Coccinellidae, Curculionidae, Dytiscidae, Elateridae, and Staphylinidae (Table 1). The percentages of each family also show similar values and arrangements in both peatlands.

The local abundance estimated for all recorded coleopteran species shows the following result in 2023: 25 coleopteran species (8.9%) were considered as abundant, 55 species (19.6%) were estimated as frequent, 89 species (31.7%) as occasional, and 112 species (39.9%) were considered rare. The estimation of abundance is based on the frequency of occurrence of the adult stage only. As it is generally subjective in our case due to different methodological approaches and the totally different bionomy of the studied objects, it should reflect the real population densities in the habitat and can be used for comparison purposes. The abundance structure does not differ in the two main ecological groups of peatland Coleoptera studied: terrestrial and aquatic beetles (amphibiontic Scirtidae with aquatic larva and terrestrial imago were included in the calculation as terrestrial). Within the aquatic beetles (all sampled with the net), the abundance composition is as follows: abundant—4 spp. (8%); frequent—10 spp. (20%); occasional—16 spp. (32%); rare—20 spp. (40%). Within terrestrial beetles (sampled by different methods), the abundance composition shows a similar arrangement: abundant—21 spp. (9.1%); frequent—45 spp. (19.5%); occasional—73 spp. (31.6%); rare—92 spp. (39.8%).

We hypothesise that the structure of abundance within coleopteran assemblage (1) depends on the successional stage of a habitat and reflects the density of beetle populations; (2) can be effectively used for monitoring peatland Coleoptera using different simple sampling methods in both aquatic and terrestrial habitats. Such an estimate of species abundance should be sufficient to investigate the effects of human activities on coleopteran assemblages, and in particular, on the rewetting of degraded peatlands.

### 3.2. Peat-Bog Biotope Fidelity and Habitat Structure

The ecological relevance of coleopteran species composition becomes clearer at the species level when we interpret our entomological biodiversity in the context of species distribution in basic peatland communities (Figure 2). This implies that species are grouped according to their fidelity to a particular peat bog biotope. As peat bogs are characterised by very specific and often extreme environments, many bog insect species are thought to be highly specialised and confined to them [6]. Thus, an intact bog should provide some stenotopic insect assemblages that could be considered the species groups of indicative value.

These biotope-confined species assemblages are generally identified using the Peus’ classification, which includes four categories [6,36]: (a) *tyrphobionts*—occurring exclusively in bogs; (b) *tyrphophiles*—typical of bogs, but not strictly confined to them; (c) *tyrphoneutrals*—distributed across different habitat types; and (d) *tyrphoxenes*—immigrants that cannot survive in bogs.

*Tyrphobiontic* species were not sampled in the study area. We suggest that this may be related to the current habitat structure and vegetation in the transformed peatland, which now differs from the typical raised bog. In the latter habitat, taxa are thought to be strongly influenced by specific conditions—high soil moisture and *Sphagnum* moss dominance [6]. These two factors should apparently be combined to provide the necessary conditions for tyrphobiontic beetles, whereas in the Vittgirrensky peatland, *Sphagna* are found sporadically and the water table is very unstable.

*Tyrphophiles* were selected for each ecological group (aquatic, terrestrial, and amphibionthic). The peatland specialist species are the following 15 beetles (Table 2). These beetles constitute 5.3% of the beetle assemblage and probably represent remnants of the original raised bog communities.

Thus, despite the lack of continuous *Sphagnum* cover, some bog specialists within the tyrphophilous group can survive on a disturbed peatland, apparently due to the other ‘true’ bog plants remaining and the preservation of the residual oligotrophic peat layer. Such environments could be considered as refugia for bog-specialised insect species, which can play a role of bioindicators within different scenarios of peatland development (including bog habitat rehabilitation).

The other 266 species sampled represent elements of various secondary habitats and are mainly eurytopic generalists. Among these, 205 species (i.e., 72.9%) were considered *tyrphoneutral* and 61 species (21.7%) were considered *tyrphoxenous*. Thus, the actual fauna of the Vittgirrensky peatland mainly consists of adaptable generalists, while relict bog elements are scarcely represented, being even less species-rich than tyrphoxenous (i.e., ‘alien to peatland’) species.

As the most common species are critical to the coleopteran assemblage, future comparative analyses can assess the impact of peatland restoration on the abundance of these species. The tyrphophiles (peatland specialists) as well as the most abundant tyrphoneutral generalists can be used to monitor potential environmental impacts on degraded peatland communities in case of restoration. The most abundant non-peatland specialist species recorded in 2023 and likely to be of indicative value are 31 tyrphoneutral species listed in Table 3. Restoration measures in the Vittgirrensky peatland could have a neutral or negative effect on the abundance of these most common tyrphoneutral species.

The present-day coleopteran assemblages and their species composition are primarily associated with the structure of the current vegetation cover in the Vittgirrensky peatland. Although 27 ‘true’ plant species are now common in the peatland [28], many of them are sporadic or do not form typical bog communities as dominant species [29]. Moreover, the composition, spatial heterogeneity, and distribution patterns of present-day peatland communities differ from those of a typical raised bog. The comparison shows that most communities on a disturbed peatland, which represent the successional stages of spontaneous vegetation recovery [29], cannot be transferred to the communities of a natural bog (Table 4). The main characteristics of disturbed peatlands are the absence of *Sphagnum* carpets in the ground layer and the small area of open sites [28]. These transformations are mainly caused by strong drainage which also leads to the loss of ‘true’ mire species and the formation of new habitats—more similar to non-bog communities (Figure 3).

All these changes result in alterations in the beetle fauna. The current coleopteran assemblage of the Vittgirrensky peatland is diverse and shows a mixture of components of different habitats and communities: edges of small-leaved forest, open habitats, secondary wetlands and ruderal communities within the agricultural landscape. Most of the coleopteran fauna of the Vittgirrensky peatland (266 spp., 94.7%) does not belong to the natural fauna of undisturbed raised bogs.

The possible rewetting of the area aims to raise the local water table and could therefore reestablish niches for species adapted to wet conditions and open water, as well as for species sensitive to significant water table fluctuations. An increase in peatland-specialised populations (15 species mentioned above) can be expected in the case of peatland restoration, but this should be factually confirmed and recorded for different taxonomic and ecological groups. Neutral effect, population decline, or spatial redistribution for non-peatland specialised species (31 observable species listed above) can be assumed but is not easy to predict. Some predictions have been made in the relevant section for each ecological group.

The comparison of numbers and percentages is relatively rough, formal, and its results can be interpreted in many different ways. Table 1 shows that the beetles of the family Staphylinidae are underrepresented in the materials collected in the Vittgirrensky peatland. This fact should be explained by the low representation of the small-sized dwellers of moose and soil in our 2023 samples. The ‘better’ representation of Cantharidae, Carabidae, Curculionidae, and Elateridae in the Vittgirrensky peatland than in the nearly intact Zehlau bog could be explained by the additional number of non-specialised species from the surrounding habitats in the ditched, drained, and heavily anthropogenically altered peatland compared to a pristine *Sphagnum* bog.

### 3.3. Ecological Characteristics and Peatland Habitat Affinity

The beetle assemblage has a mixed character and consists of wetland-specialised species, forest species, hydrophilous near-water species, and species associated with open landscapes. The beetle assemblage of the Vittgirrensky peatland can be divided into three main ecological groups, which will be discussed separately: aquatic beetles (seven families: Hydrophilidae, Hydraenidae, Hydrochidae, Dytiscidae, Haliplidae, Gyrinidae, and Noteridae), amphibiontic beetles (single family Scirtidae with aquatic larva and terrestrial imago), and terrestrial beetles (33 families, with only a few representatives associated with peat soils, mire plants, banks of waterbodies). The latter group is further divided into subgroups (see below).

#### 3.3.1. Aquatic Beetles

Based on Ryndevich [34], all the water beetles of the geographically and climatically close Belarus can be divided into several groups according to their preference for flowing or stagnant waterbodies (reophilic, reobiontic, stagnobiontic) and their preference for water pH values (acidophilic, eurytopic, and alcalophilic). The water beetle assemblage of the Vittgirrensky peatland (50 spp. in total) includes three groups of the classification (Table 5): reophilic (inhabitants of the both flowing and stagnant waters; 23 spp., 46%), eurytopic stagnobionts (inhabitants of stagnant waters without pH preference; 25 spp., 50%) and acidophilic stagnobionts (inhabitants of stagnant waters with pH values ranging 3.3–6.8; two spp., 4%). Many species of the first two groups also occur in pristine peat bogs and have been reported, e.g., from the Zehlau bog [9,23], but these inhabitants are not bog specialists and could repeatedly enter the study area by air or even directly from neighbouring water bodies without leaving the water.

The most ecologically interesting group is that of the acidophilic stagnobionts, which prefer acidic waters and are represented in the study area by *Hydrochus elongatus* and *Enochrus ochropterus*. These two species were sampled in the southern part of the Vittgirrensky peatland in a non-drying drainage ditch, where the most diverse water assemblage was recorded in 2023. It is likely that these species only used this water body as a temporary refuge when their preferred water bodies dried up, and that the high diversity here is a consequence of the secondary concentration of flying adults.

The most abundant and widespread species of the Vittgirrensky peatland in 2023 are *Haliplus ruficollis* (reophilic), *Hyphydrus ovatus* (reophilic), and *Anacaena lutescens* (eurytopic stagnobiont); other still common and widespread species are *Acilius canaliculatus* (reophilic), *Hygrotus inaequalis* (reophilic), *Hydroporus angustatus* (eurytopic stagnobiont), and *H. neglectus* (eurytopic stagnobiont).

We suggest that the further monitoring of the Vittgirrensky peatland should pay more attention to indicator aquatic species (such as acidophilic stagnobionts), but changes in the water beetle assemblage may also occur in the distribution or abundance of other community members. It can be hypothesised that the restoration measures in the Vittgirrensky peatland will lead to an increase in the water level and the number of permanent water bodies in the area. The increase in stagnobiont (and specialised acidophilic stagnobionts) occurrence and population density in such cases could be predicted for the central area of the peatland. The reophilic elements would favour larger drainage ditches in the periphery of the peatland, where their diversity and abundance might increase. The general zonation and ecological stratification must take place.

#### 3.3.2. Amphibiontic Beetles

This group of beetles is represented by five species of the single family Scirtidae: *Contacyphon ochraceus*, *C. padi*, *C. variabilis*, *Microcara testacea*, and *Scirtes hemisphaericus*. Three of these species are considered to be common in the study area in 2023: *Contacyphon padi*, *C. variabilis*, and *Scirtes hemisphaericus*. All adult scirtids are associated with shoreline habitats, and *Contacyphon variabilis* is a common species of peatlands and raised bogs. The likely rise in water levels and the increase in the number of permanent water bodies in the Vittgirrensky peatland after restoration should lead to an increase in the abundance and probably species richness of these beetles due to the formation of additional larval development sites, such as small persistent puddles and ditches.

#### 3.3.3. Terrestrial Beetles

This ecological group (33 families, 226 species) is the most diverse in terms of species and bionomy, and therefore the most difficult to analyse. For the purposes of analysis, we divide the sampled terrestrial beetles into three nominal ecological groups (Table 6), such as plant-associated phytophagous beetles, epigeic elements, and ‘miscellaneous’ Coleoptera (i.e., other families including representatives of mycetobionts, predators, pallinophagous vegetation feeders, etc.). The subgroups of the terrestrial beetle assemblage of the Vittgirrensky peatland are briefly and separately discussed below.

#### 3.3.4. Miscellaneous Terrestrial Beetles

Many of the beetles in this subgroup are not directly associated with peatland (or even wetland) habitats in the adult or in all stages and were recorded in the study area as rare or occasional residents or short-distance migrants in a specific ‘ruderal’ community of the central roadside and shrub vegetation of peatland margins. The subgroup includes 17 families (Table 6). Among them, Cantharidae, Coccinellidae, and Malachiidae are predaceous or pallinophagous on vegetation. Two species of these families collected in 2023 are considered to be specific to peatlands: *Scymnus suturalis and Malthodes pumilus*. Other recorded representatives are eurytopic generalists and are associated with a wide range of non-specific semi-open habitats (shrubs, forest margins, etc.). Members of Cerambycidae, Elateridae, Lucanidae, Mordellidae, Oedemeridae, Pyrochroidae, Scarabaeidae (Melolonthinae, Cetoniinae), Scraptiidae, Tenebrionidae, and Throscidae are usually the hidden inhabitants of plant tissues (mainly wood) or soil in the larval stage and visitors of vegetation (often flowers) in the adult stage. The arboreal vegetation of the Vittgirrensky peatland (*Betula*, *Populus*, *Salix*, *Pinus*) and the presence of decayed wood determine the fauna of these beetles. Lucanidae are not considered to be part of the local beetle community, the single observed specimen of *Dorcus* represents a tourist (i.e., not resident) species in the peatland.

Restoration measures in the peatland, resulting in water level rises and vegetation changes in the core peatland area, are unlikely to have a major impact on the populations on most of these insects. The relatively rare species *Oedemera croceicollis* may be positively affected by the rewetting of the area. Two registered members of the dung community (Scarabaeidae: Aphodiinae) are non-specialised and are associated with open landscapes. Mycetobiontic Leiodidae (e.g., *Agathidium*), Cryptophagidae, Endomychidae, and Latridiidae are associated with decaying plant material and various fungi. Also, for this ecological group of Coleoptera, no negative effects of the restoration measures in the Vittgirrensky peatland are predicted. Apparently, many species of this group have a high dispersal and re-colonisation ability and can easily invade suitable habitats from the outside. Several additional representatives of Monotomidae, Nitidulidae, Ciidae, and Scolytinae not recorded in 2023 are expected to be found in the periphery of the peatland during targeted searches in the closed canopy birch forest.

It is noteworthy that one of the species recorded in the Vittgirrensky peatland (along the central road) in 2023, *Harmonia axyridis*, is a recent (first decade of the XXI century) active invader in the territory of Kaliningrad Oblast. This species is now common in towns and cities, and probably arrived in the Vittgirrensky peatland by transport after the resumption of human activity in the area in recent years.

#### 3.3.5. Epigeic Beetles

Epigeic beetles include the representatives of five families: Byrrhidae, Carabidae, Dermestidae, Silphidae, and Staphylinidae. The composition of this group is nominal because the family Staphylinidae is very diverse in its bionomy and includes mycetobionts, predators occurring on soil, fungi and vegetation, scavengers, inquilines of ant nests, etc. Perhaps the most important unifying feature of the five families mentioned above is the method of collection: the beetles of this group are effectively sampled using pitfall traps. The majority of the sampled epigeic beetles are not bog or peatland specialists and can be characterised as open-habitat species (e.g., *Amara famelica*, *Cymindis vaporariorum*, *Drusilla canaliculata*, *Poecilus cupreus*, *Stenus solutus*), forest species (e.g., *Anthobium unicolor*, *Carabus nemoralis*, *Leistus terminatus*, *Pterostichus niger*, *Sepedophilus testaceus*, *Xantholinus longiventris*), or necrophagous species (e.g., *Nicrophorus vespilloides*). Two of the most characteristic bog species in the Baltic Region, *Agonum ericeti* and *Pterostichus rhaeticus* [11,37], were not found in the Vittgirrensky peatland.

The most abundant and widespread epigeic species of the Vittgirrensky peatland in 2023 are *Acupalpus brunnipes*, *Anthobium unicolor*, *Apocatops nigrita*, *Leistus terminatus*, *Nicrophorus vespilloides*, *Pterostichus niger*, *Sciodrepoides watsoni*, *Tachyporus dispar*, and *Xantholinus longiventris*. Common and abundant are *Amara famelica*, *Poecilus cupreus*, *Acidota crenata*, *Drusilla canaliculata*, *Oxypoda opaca*, *Philonthus cognatus* and *Platydracus latebricola*. *Proteinus brachypterus* and *Stenus solutus* are also widespread and probably ubiquitous in the area, but are considered to be occasional or rare due to the small number of recorded specimens.

The epigeic beetle assemblage of the study area is diverse and includes a mixture of fauna from different habitats. The following epigeic coleoptera can be considered as peatland specialists: *Acupalpus brunnipes*, *Bradycellus rufcollis*, *Curimopsis nigrita*, *Platydracus fulvipes,* and *P. latebricola*. In addition, species such as *Brachygluta haematica*, *Ischnosoma longicorne*, *Pselaphus heisei* are more characteristic of peatlands and prefer similar habitats, but are not strictly stenotopic (e.g., *P. heisei* also occurs on forest edges). A number of species are associated with different types of riparian zones, inhabiting the margins of drainage ditches or *Phragmites*-dominated communities: *Bembidion quadrimaculatum*, *Dyschirius globosus*, *Hygronoma dimidiata*, *Lesteva longoelytrata*, *Pterostichus anthracinus*, *Stenus* spp. Representatives of Dermestidae, Silphidae, and Leiodidae (Cholevinae) are almost exclusively members of necrophagous communities and are therefore not true peatland specialists. Restoration measures may negatively affect the populations of the sun-adapted species of open habitats and dry conditions (*Cymindis vaporariorum*, *Cicindela campestris*), but will favour the hydrophilous beetle assemblage of shores and wetland communities, e.g., associated with moss cover.

It should be noted that wetland specialised species in the Vittgirrensky peatland are much more numerous and abundant among epigeic beetles than among aquatic ones. It can be assumed that the degradation of the aquatic coleopteran community in the study area is deeper, while the terrestrial coleopteran communities have time to recover in the period after the end of peat extraction. The natural water bodies of the raised bog have almost completely disappeared, whereas several microhabitats of terrestrial epigeic beetles remain in small patches or partially.

#### 3.3.6. Plant-Associated Phytophagous Beetles

All representatives of nine families were considered plant-associated phytophagous beetles: Attelabidae, Brentidae, Byturidae, Chrysomelidae, Curculionidae, Kateretidae, Megalopodidae, Nitidulidae, and Phalacridae. In addition, three species associated with living vegetation from Buprestidae (*Trachys minutus*), Cerambycidae (*Agapanthia villosoviridescens*) and Cryptophagidae (*Telmatophilus typhae*) are considered phytophagous and are included in this subgroup, which consists of 73 species from 12 families. Among these beetles, 8 species were considered abundant, 11 species were considered frequent, 23 species are occasional, and 31 species are rare. Most of the abundant and frequent species are associated with *Betula* and *Salix* (*Altica aenescens*, *Lochmaea caprea*, *Polydrusus picus*, *Crepidodera aurata*, *Orchestes jota*, etc.) or ruderal plants of the central road such as *Urtica*, *Chamaenerion*, *Rubus*, *Cirsium*, and *Carduus* (*Brachypterus urticae*, Neocrepidodera transversa, etc.). Only one ‘abundant’ species, *Altica longicollis*, is associated with the peatland (communities with *Calluna*) and only one frequent species, *Telmatophilus typhae*, indicates the wetland area with the *Typha* community.

From the compiled list of phytophagous beetles (Table 7), only two species, *Altica longicollis* and *Micrelus ericae*, can be considered to be peatland specialists in the study area, as they are trophically related to the peatland dominant *Calluna vulgaris*. Many other species are often found in different types of wetlands, although they tend to be eurytopic and mainly depend on the distribution of fodder plants (Table 7). The distribution of these beetles should generally coincide with the distribution of the fodder plants in the Vittgirrensky peatland and can be predicted using vegetation maps.

It is interesting to note that representatives of *Kateretes* (Kateretidae), Donaciinae (Chrysomelidae), *Limnobaris,* and Erirhinini (Curculionidae), which are characteristic of wetland and associated with diverse plant communities, including their forage plants *Typha*, *Phragmites*, *Scirpus*, *Carex*, *Sparganium*, etc., were not sampled in the study area. This fact may be related to the strong seasonal fluctuations of the water table in the area and the drying up of the water bodies. These beetles were expected to be found on the northern edge of the peatland. The colonisation of the central area by these beetles after rewetting of the peatland is also likely.

### 3.4. Note

Several collection methods that are effective in wetlands were not used in our surveys (e.g., the light trapping and sieving of substrates). Pitfall trapping was only carried out at three sampling sites. The resulting species list has a preliminary character and reflects the beetle assemblage of mainly central, western, and southern areas of the Vittgirrensky peatland, and is rather incomplete with respect to small-sized soil dwellers. The possible inter-annual variation of the beetle assemblage cannot be estimated from the single six-month survey. Taking into account the fact that both spatial and temporal factors influence the invertebrate faunistic diversity of peatlands, further studies in the Vittgirrensky peatland may add a number of additional species to the list of inhabitants and correct the estimates of species abundance.

The long-term monitoring and data collection of Coleoptera is recommended and may be of interest in any case of the possible management of the Vittgirrensky peatland, i.e., rewetting by damming ditches or leaving the situation unchanged. In spite of all the difficulties mentioned, the data obtained in 2023 provide baseline information for comparison and monitoring of environmental changes in the case of the rewetting of the Vittgirrensky peatland and can be used in before–after (BA) or even in before–after control-impact (BACI) studies.

## 4. Conclusions

After more than 20 years of abandonment and the spontaneous re-establishment of vegetation cover on a cut-over peatland (Vittgirrensky) in the Kaliningrad region, a total of 281 coleopteran species from 41 families were recorded here from different sites. This beetle assemblage is considered to be a pre-impact fauna to be further investigated during the ‘before–after’ (BA) or ‘before–after control-impact’ (BACI) study on a peatland which is planned to be rewetted.

Compared to an intact raised bog, the Vittgirrensky peatland has a less specialised beetle assemblage, showing an intermixed fauna of small-leaved forest edges, open meadows, wetlands, and ruderal communities of the agricultural landscape. Tyrphobiontic species (bog specialists) are completely absent from the peatland, while tyrphophiles (typical of bogs) still constitute 5.3% (15 spp.) of the beetle fauna, probably representing the remnants of the former raised bog communities. The major part of today’s peatland coleopteran assemblage are tyrphoneutral (205 spp., 72.9%), which belong to adaptable generalists from various non-bog habitats.

The current coleopteran species composition of the abandoned disturbed peatland is associated with the structure of its current vegetation cover, which is characterised by a fragmentary distribution of *Sphagnum* carpets, lack of open habitats, and widespread birch coppice or tree stands. Most of today’s peatland communities do not correspond to those of a typical European raised bog.

The heterogeneity of vegetation and soil cover determines the diversity of coleopteran ecological groups in the peatland, with groups such as terrestrial (226 spp., including plant-associated phytophages, epigeic, ‘miscellaneous’), aquatic (50 spp., including reophilic species, eurytopic stagnobionts, acidophilic stagnobionts), and amphibiontic (5 spp.).

The rewetting measures will most likely have a positive effect on the stenotopic peatland-specialised group of tyrphophiles (15 spp.) and—but neutral or negative (populations decrease or spatial redistribution)—on a group of the most abundant tyrphoneutral generalists (31 spp.). These two coleopteran aggregates of 46 species can be considered as an indicator complex (Figure 4) used to monitor environmental changes during further peatland development in terms of different scenarios.

## Figures and Tables

**Figure 1 insects-15-00356-f001:**
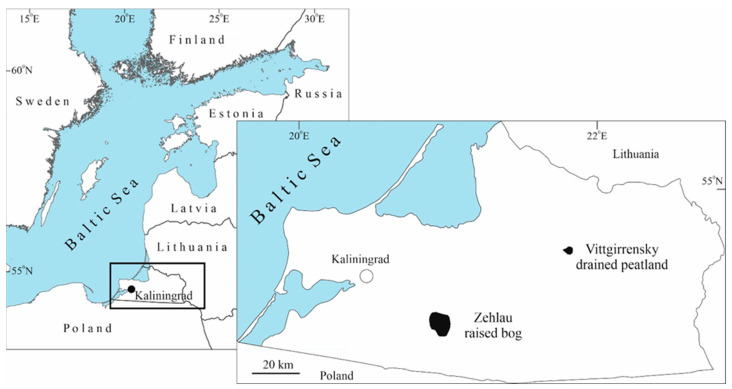
Location of the Vittgirrensky peatland (the ‘Rossyanka’ Carbon Measurement Supersite) and the Zehlau raised bog (mentioned in the text for comparison) in Kaliningrad Oblast.

**Figure 2 insects-15-00356-f002:**
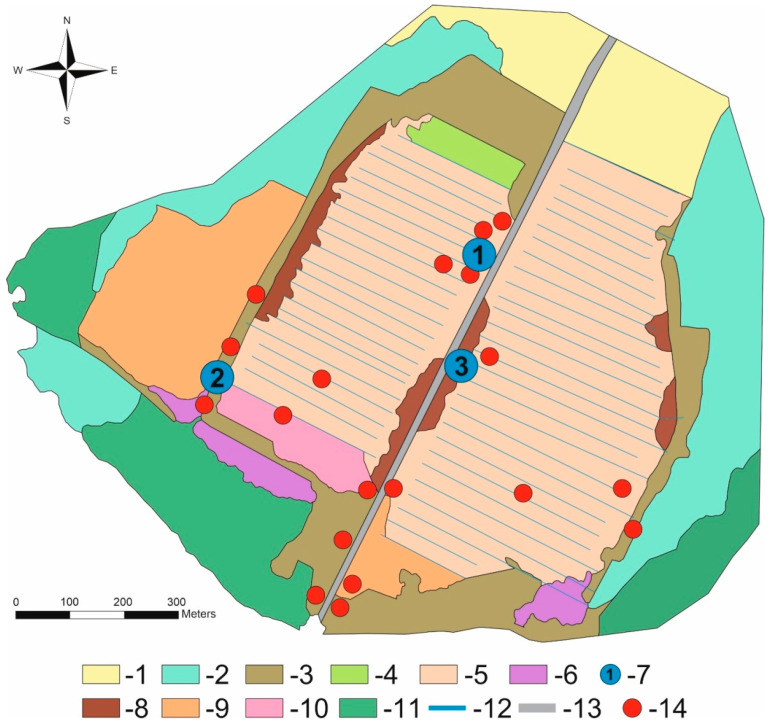
Vegetation cover in the Vittgirrensky Peatland (after [28,29]) and location of the sampling points for Coleoptera in 2023: 1—Dry shrublands, 2—Wet shrublands, 3—Dry birch stand, 4—Fen-like communities (*Juncus*), 5—Birch coppice, 6—Reed beds, 7—pitfall traps set in lines, 8—Bare-peat sites, 9—Dense closed-canopy stand, 10—Fen-like communities (*Eriophorum*/*Carex*), 11—Wet forest (birch and aspen), 12—Hydrophilic communities in ditches, 13—Dirt road, 14—Places of sweeping with net in water.

**Figure 3 insects-15-00356-f003:**
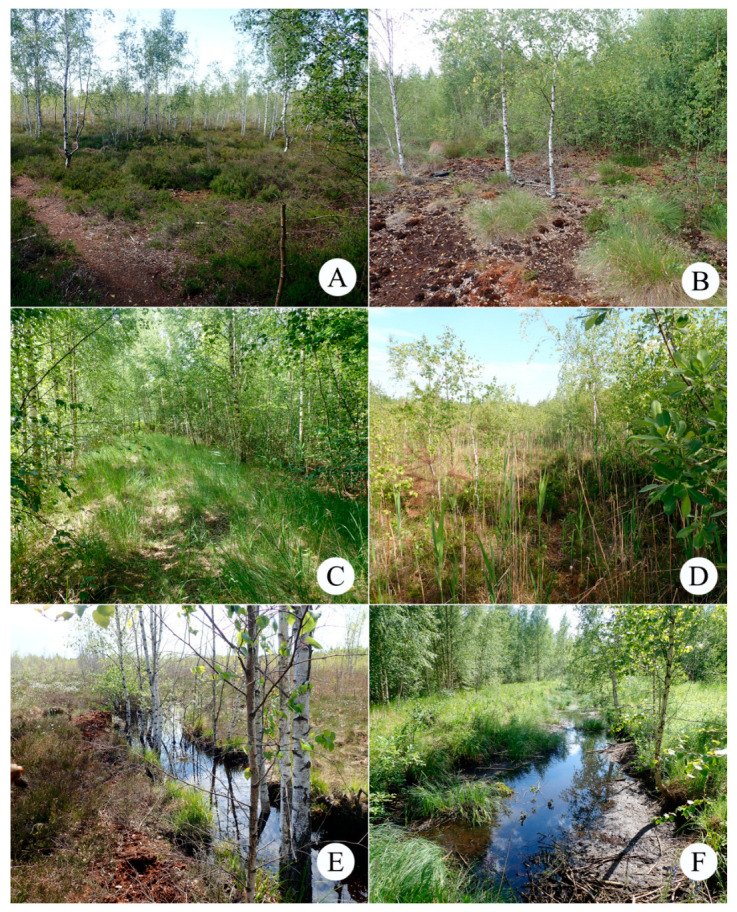
The main typical habitats of the Vittgirrensky peatland: (**A**) Birch coppice with *Calluna* (site of the pitfall line 1), habitat for *Micrelus ericae*, *Curimopsis nigrita*, *Altica longicollis*; (**B**) Open bare peat (site of the pitfall line 3), habitat for *Cymindis vaporariorum*, *Parabolitobius formosus*, *Pselaphus heisei*, *Curimopsis nigrita*; (**C**) Dry birch stand (site of pitfall line 2), habitat for *Sciaphilus asperatus*, *Barypeithes pellucidus*, *Brachysomus echinatus*; (**D**) *Phragmites*-dominated birch coppice, habitat for *Malthodes pumilus*, *Scymnus suturalis*; (**E**) The drainage ditch drying up in summer, sampling place of *Dytiscus dimidiatus*, *Graphoderus cinereus*, *Hydaticus seminiger*; (**F**) non-drying drainage ditch, sampling place of *Hydrochus elongatus*, *Enochrus ochropterus*.

**Figure 4 insects-15-00356-f004:**
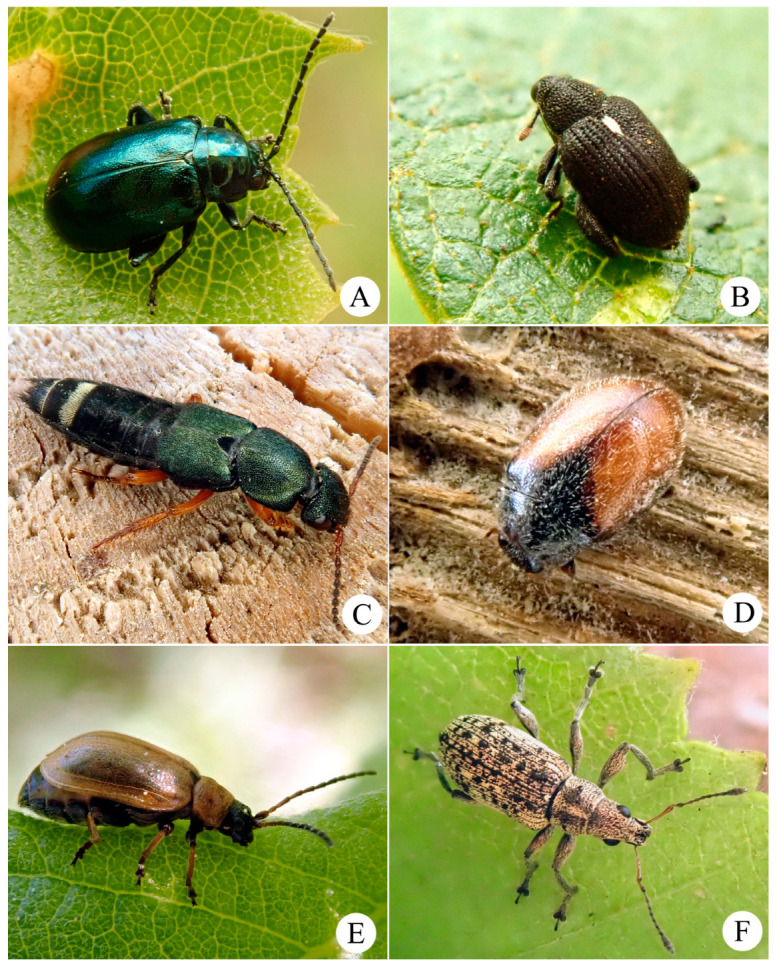
Some of the potential indicator species (tyrphophiles and abundant tyrphoneutrals) that can be used, among others, to monitor environmental impacts in the Vittgirrensky peatland: (**A**) *Altica aenescens*; (**B**) *Orchestes jota*; (**C**) *Platydracus fulvipes*; (**D**) *Scymnus suturalis*; (**E**) *Lochmaea caprea*; and (**F**) *Polydrusus cervinus*.

**Table 1 insects-15-00356-t001:** The most species-rich families (≥10 species) in the Vittgirrensky peatland (own results) and the Zehlau bog [9,11,23].

No	Family	Vittgirrensky		Zehlau	
		Number of Species	%	Number of Species	%
1	Cantharidae	13	4.6	10	2.6
2	Carabidae	30	10.7	36	9.3
3	Chrysomelidae	24	8.5	41	10.6
4	Coccinellidae	10	3.6	20	5.2
5	Curculionidae	30	10.7	38	9.8
6	Dytiscidae	28	10.0	43	11.1
7	Elateridae	11	3.9	11	2.8
8	Staphylinidae	43	15.3	85	22.0
	Coleoptera in total	281	100	387	100

**Table 2 insects-15-00356-t002:** Tyrphophile beetle species of the Vittgirrensky peatland.

Peatland Habitat and Ecological Affinity	Tyrphophile Species
Aquatic acidophilic stagnobionts	*Enochrus ochropterus*, *Hydrochus elongatus*
Amphibiontic beetles	*Contacyphon variabilis*
Terrestrial dwarf shrub beetles	*Malthodes pumilus*, *Scymnus suturalis*
Epigeic beetles	*Acupalpus brunnipes*, *Brachygluta haematica*, *Bradycellus ruficollis*, *Curimopsis nigrita*, *Ischnosoma longicorne*, *Platydracus fulvipes*, *P. latebricola*, *Pselaphus heisei*
Phytophagous beetles	*Altica longicollis*, *Micrelus ericae*

**Table 3 insects-15-00356-t003:** The most abundant non-peatland specialists (tyrphoneutral species) within the coleopteran assemblage of the Vittgirrensky peatland.

Habitat Affinity	Species
Aquatic environment	*Haliplus ruficollis*, *Hyphydrus ovatus*, *Anacaena lutescens*, *Acilius canaliculatus*, *Hygrotus inaequalis*, *Hydroporus angustatus*, *H. neglectus*
Epigeic assemblage	*Anthobium unicolor*, *Apocatops nigrita*, *Leistus terminatus*, *Nicrophorus vespilloides*, *Pterostichus niger*, *Sciodrepoides watsoni*, *Tachyporus dispar*, *Xantholinus longiventris*, *Amara famelica*, *Poecilus cupreus*, *Acidota crenata*, *Drusilla canaliculata*, *Oxypoda opaca*, *Philonthus cognatus*
Plant-associated beetles	*Altica aenescens*, *Lochmaea caprea*, *Brachysomus echinatus*, *Polydrusus picus*, *Telmatophilus typhae*, *Bromius obscurus*, *Crepidodera aurata*, *Orchestes jota*, *Polydrusus cervinus*, *Sciaphilus asperatus*

**Table 4 insects-15-00356-t004:** Comparison of basic plant communities on an idealised Central European raised bog and in the Vittgirrensky peatland.

Typical Central European Raised Bog (after [6])	Vittgirrensky Peatland [28]
(1) Central bog pool	Hydrophilic communities in ditches (!)—only separate fragments and only for a limited period of time
(2) Pine elfin forest	Absent
(3) Open dead pine forest and treeless formation with high water level	Absent
(4) Birches (*Betula* spp.) and pines (*Pinus* spp.) on inner lagg (moat) margin	Birch coppice (!)—only separate small fragments with Sphagna
(5) Treeless lagg	Fen-like communities (!)—only for a limited period of time
(6) Border communities between lagg and surrounding forest	Absent
(7) Surrounding forest	- Wet forest (birch and aspen) - Dry birch stand - Dense closed-canopy stand
Absent	Birch coppice (most part without Sphagna)
Absent	Dry shrublands
Absent	Wet shrublands
Absent	Reed beds
Absent	Bare-peat sites

**Table 5 insects-15-00356-t005:** Water beetle assemblage of the Vittgirrensky peatland.

Reophilic Species	Eurytopic Stagnobionts	Acidophilic Stagnobionts
*Gyrinus substriatus*, *Haliplus immaculatus*, *H. lineatocollis*, *H. ruficollis*, *P. caesus*, *Noterus clavicornis*, *Acilius canaliculatus*, *A. sulcatus*, *Agabus bipustulatus*, *Dytiscus dimidiatus*, *Graphoderus cinereus*, *Graptodytes pictus*, *Hydaticus seminiger*, *Hydroporus obscurus*, *H. palustris*, *Hygrotus impressopunctatus*, *H. inaequalis*, *Hyphydrus ovatus*, *Ilybius ater*, *Laccophilus minutus*, *Porhydrus lineatus*, *Rhantus suturalis*, *Scarodytes halensis*	*Haliplus fulvus*, *Clemnius decorates*, *Graptodytes granularis*, *Hydaticus transversalis*, *Hydroporus angustatus*, *H. dorsalis*, *H. erythrocephalus*, *H. glabriusculus*, *H. neglectus*, *H. planus*, *Ilybius guttiger*, *Rhantus exsoletus*, *Helophorus aquaticus*, *H. granularis*, *H. flavipes*, *Anacaena lutescens*, *Cymbiodyta marginella*, *Enochrus coarctatus*, *E. testaceus*, *Helochares obscurus*, *Hydrobius fuscipes*, *Hydrochara caraboides*, *Laccobius minutus*, *Limnebius crinifer*, *Ochthebius minimus*	*Enochrus ochropterus*, *Hydrochus elongatus*

**Table 6 insects-15-00356-t006:** Subgroups of the terrestrial beetle assemblage in the Vittgirrensky peatland.

Plant-Associated Phytophagous Beetles	Epigeic Elements	‘Miscellaneous’ Coleoptera
Attelabidae, Brentidae, Byturidae, Buprestidae, Chrysomelidae, Curculionidae, Kateretidae, Megalopodidae, Nitidulidae, Phalacridae	Byrrhidae, Carabidae, Staphylinidae, necrobiontic Dermestidae, Silphidae, Leiodidae: Cholevinae	Cantharidae, Cerambycidae, Coccinellidae, Cryptophagidae, Elateridae, Endomychidae, Latridiidae, Leiodidae, Lucanidae, Malachiidae, Mordellidae, Oedemeridae, Pyrochroidae, Scarabaeidae, Scraptiidae, Tenebrionidae, Throscidae

**Table 7 insects-15-00356-t007:** Fodder plant association among phytophagous beetles in the Vittgirrensky peatland.

Fodder Plants	Phytophagous Beetle Species
*Calluna vulgaris*	*Altica longicollis* and *Micrelus ericae*
*Betula* spp.	*Altica aenescens*, *Betulapion simile*, *Orchestes jota*, *Phyllobius argentatus*, *Polydrusus cervinus*
*Salix* spp.	*Archarius salicivorus*, *Byctiscus betulae*, *Crepidodera aurata*, *Cryptocephalus decemmaculatus*, *Cryptocephalus ocellatus*, *Pachybrachis hieroglyphicus*, *Polydrusus picus*, *Temnocerus coeruleus*, *Trachys minutus*
*Salix* spp. and *Betula* spp.	*Lochmaea caprea*, *Rhamphus pulicarius*
*Populus tremula*	*Byctiscus populi*, *Chrysomela populi*, *Zeugophora subspinosa*
*Alnus glutinosa*	*Agelastica alni*
Polygophagous on Cruciferae	*Brassicogethes viridescens*, *Phyllotreta atra*, *Phyllotreta striolata*
Polyphagous on arboreal plants	*Phyllobius pyri*
Cyperaceae	*Phalacrus championi*
Gramineae	*Oulema obscura*
*Lythrum salicaria*	*Nanophyes marmoratus*
*Typha latifolia*	*Telmatophilus typhae*

## Data Availability

All relevant datasets in this study are described in the manuscript.

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
