# Peer review of "Ecological Groups of Coleoptera (Insecta) as Indicators of Habitat Transformation on Drained and Rewetted Peatlands: A Baseline Study from a Carbon Supersite, Kaliningrad, Russia"

_insects, 2024, doi:10.3390/insects15050356_

Round 1

Reviewer 1 Report

Comments and Suggestions for Authors

1. It is necessary to clarify the volume of the collected material in the Chapter "2. Materials and Methods".

2. There is no data on abundance in the Chapter "3.1. Taxonomic structure and abundance". It is advisable to provide data on the number of particular taxa in different biotopes based on collection using entomological sweep nets or pitfall traps.

Author Response

Dear Reviewer,

Thank you very much for your comments and suggestions.

We have made some improvements to the text according to your opinion.

Reviewer 2 Report

Comments and Suggestions for Authors

The manuscript is very interesting, especially for the number of species collected.

The wording of the methodology is confusing. More details should be given.

It is always very important to go to expert taxonomists in each group to corroborate the identifications.

A single person can make identification errors, especially with specimens from so many different families of beetles.

All scientific names lack author and year of identification. They should be mentioned every time the scientific name of a species is written for the first time.

Author Response

(The authors gave the same response as above.)

Reviewer 3 Report

Comments and Suggestions for Authors

Dear authors,

The manuscript deals in an interesting topic of insect communities in a wetland habitat which is going to be more and more important in the future. Although it is just a preliminary experiment, and the conditions of the insect fauna is going to be followed in the future, the authors obtained significant results regarding the number of Coleopteran species identified at the studied site.

The introduction in very well written, clear and concise, with defined objectives and importance of the topic. Material and methods are described well enough that the study can be replicated. The methods themself are sound and based in literature sources. Results and discussion section are written wide and comprehensively, but the problem is that some parts are just located in the wrong places (results in the discussion and discussion in the results, and some parts of methodology in the discussion). That is the only reason why I suggest major revision. So, the results referring to the affiliation to habitat preference should be in the results section, and all the other information regarding interpretation of those results, explanation why it is as such, and comparison to the other studies’ results should go to the discussion section. As the journal accepts free format submission, I would suggest combining Results and Discussion sections into one (Results and discussion), and shortening it a little.  Also, listing of more than 5 species in a sentence is burdening to the manuscript, and makes it harder to read it. If the author could condense those listings into one ore more tables, it would make the manuscript more appealing to the reader, and the results could be more comparable when reading. The point by point comments are listed in the attached file.

Author Response

(The authors gave the same response as above.)

Round 2

Reviewer 3 Report

Comments and Suggestions for Authors

Dear authors,

You have improved the quality of the manuscript, so I suggest accepting it in this form.